

# Symbiotic immuno-suppression: is disease susceptibility the price of bleaching resistance?

Daniel G. Merselis[1], Diego Lirman[2] and Mauricio Rodriguez-Lanetty[1]

[1] Department of Biological Sciences, Florida International University, Miami, FL, USA
[2] Department of Marine Biology and Ecology, University of Miami, Miami, FL, USA

## ABSTRACT

Accelerating anthropogenic climate change threatens to destroy coral reefs worldwide through the processes of bleaching and disease. These major contributors to coral mortality are both closely linked with thermal stress intensified by anthropogenic climate change. Disease outbreaks typically follow bleaching events, but a direct positive linkage between bleaching and disease has been debated. By tracking 152 individual coral ramets through the 2014 mass bleaching in a South Florida coral restoration nursery, we revealed a highly significant negative correlation between bleaching and disease in the Caribbean staghorn coral, *Acropora cervicornis*. To explain these results, we propose a mechanism for transient immunological protection through coral bleaching: removal of *Symbiodinium* during bleaching may also temporarily eliminate suppressive symbiont modulation of host immunological function. We contextualize this hypothesis within an ecological perspective in order to generate testable predictions for future investigation.

## INTRODUCTION

Anthropogenic climate change threatens to destroy coral reefs globally before the end of the century (*Hoegh-Guldberg et al., 2007*; *Hoegh-Guldberg, 2014*). Increasing frequency, severity, and duration of thermal anomalies have caused increased coral bleaching and disease outbreaks (*Harvell et al., 1999*; *Harvell et al., 2002*; *Bruno et al., 2007*; *Hoegh-Guldberg & Bruno, 2010*; *Ruiz-Moreno et al., 2012*; *Randall & Van Woesik, 2015*). Coral bleaching represents the breakdown of the obligate mutualism between dinoflagellates of the genus *Symbiodinium* and reef building corals. This breakdown results in decreased coral growth, fecundity, and survivorship, as the loss of photosynthetic *Symbiodinium* deprives corals of up to 95% of their energetic budget (*Muscatine & Porter, 1977*; *Glynn, 1983*; *Harriott, 1985*; *Goreau & Macfarlane , 1990*; *Szmant & Gassman, 1990*; *Baird & Marshall, 2002*). Coral tissue-loss disease outbreaks frequently follow bleaching events (*Harvell et al., 2001*; *Muller et al., 2008*; *Brandt & McManus, 2009*; *Cróquer & Weil, 2009*; *Miller et al., 2009*; *Precht et al., 2016*; *Lewis et al., 2017*) and, like bleaching, are linked to thermal anomalies (*Selig et al., 2006*; *Bruno et al., 2007*; *Brandt & McManus, 2009*; *Cróquer & Weil, 2009*; *Ban, Graham & Connolly, 2012*; *Ruiz-Moreno et al., 2012*) as well as poor water

Corresponding author
Mauricio Rodriguez-Lanetty,
rodmauri@fiu.edu

quality (*Haapkylä et al., 2011*; *Vega Thurber et al., 2014*). Many of these diseases remain poorly characterized and may represent the invasion of one or more opportunistic microbes or viruses (see *Lesser et al., 2007*; *Bourne et al., 2009*). Koch's postulates have been fulfilled for several coral diseases, but some of these same diseases have later been induced by alternative etiological agents, indicating that signs of coral maladies may constitute syndromes with many potential pathologies rather than a singular pathology (*Denner et al., 2003*; *Lesser et al., 2007*; *Sunagawa et al., 2009*; *Sutherland et al., 2011*; *Lesser & Jarett, 2014*). Like bleaching, coral tissue loss diseases can cause coral mortality, reduce coral growth and fecundity, and are recognized as major drivers of coral reef decline (*Richardson et al., 1998*; *Harvell et al., 2001*; *Patterson et al., 2002*; *Miller et al., 2006*; *Weil, Cróquer & Urreiztieta, 2009*; *Miller et al., 2009*). Coral tissue loss diseases (as opposed to diseases resultant in discoloration or abnormal growth form) are the focus of this study.

Whether tissue loss disease outbreaks follow bleaching events on a correlational or causal basis is a topic of debate (*Bruno et al., 2007*; *Muller et al., 2008*; *Brandt & McManus, 2009*; *Cróquer & Weil, 2009*; *Ban, Graham & Connolly, 2012*). A causal relationship between the two conditions is intuitive as starvation induced by bleaching could lead towards increased coral host susceptibility. *Muller et al. (2008)* and others demonstrated that a relationship between temperature and disease prevalence could be found during a bleaching year as opposed to non-bleaching years and further, that mortality due to disease was correlated to temperature in bleached, but not unbleached corals. Furthermore, there is a relationship between mean percentage of bleached corals and prevalence of several diseases in numerous Caribbean scleractinian genera (*Brandt & McManus, 2009*; *Cróquer & Weil, 2009*). These relationships correlate bleaching and disease, but do not necessarily link them mechanistically. The co-occurrence of bleaching and tissue loss diseases is expected even if the two conditions are mechanistically independent, because bleaching and tissue loss diseases are both enhanced by thermal stress (*Glynn & D'Croz, 1990*; *Bruno et al., 2007*; *Lesser, 2011*). Monitoring at the population level can indicate correlation between bleaching and disease, but cannot be used to prove a mechanistic link. A causal relationship between bleaching and disease would leave a pattern of co-occurrence when monitored at the individual level (i.e., bleached individuals should have significantly greater rates of disease). As such, monitoring efforts which perform repeated transects without tracking individuals may be unable to differentiate a causal or correlational relationship (*Cróquer & Weil, 2009*). Population and community level co-observation between bleaching and disease linked by a common environmental driver should not be construed as a dependency between them.

Contrary to this expected pattern of correlation, white band disease on the Great Barrier Reef has had a negative spatial correlation to bleaching events, even though the disease was correlated with thermal anomaly (*Bruno et al., 2007*). Further, geographically predictive models for white syndrome outbreaks are not improved by the incorporation of information known to accurately predict coral bleaching (*Ban, Graham & Connolly, 2012*). This work suggests a correlational rather than causal relationship, because these disease outbreaks are not enhanced by prior bleaching.

At a physiological level, immunological markers respond to bleaching conflictingly; prophenol oxidase and peroxidase activity may increase during bleaching, while phenol oxidase, lysozyme-like, and microbial antibacterial activity decline (*Ritchie, 2006*; *Mydlarz et al., 2009*; *Palmer, Bythell & Willis, 2011*). The coral mucus layer acts both as a physical barrier to infection and a point of first contact/adhesion for an infectious agent (*Banin et al., 2001*; *Brown & Bythell, 2005*). It is largely produced with resources from *Symbiodinium*, and its production is therefore dependent upon the mutualism between *Symbiodinium* and coral host (*Brown & Bythell, 2005*).

In the present study, monitoring for bleaching and tissue loss disease was carried out in restoration nursery. Coral nurseries provide a unique opportunity for monitoring, because histories of environmental conditions and genetic backgrounds is known in these common gardens (*Lirman & Schopmeyer, 2016*). Mother colonies are often fragmented many times, resulting in clonal individual colonies known as ramets ideal for replication. The collection of all these clonal ramets descendant from a single mother colony are known as a genet, although this is frequently referred to as a genotype in the restoration literature (*Baums, 2008*).

Individual *Acropora cervicornis* ramets were monitored during a bleaching event and subsequent recovery in an *in situ* coral nursery located near Miami, Florida, USA to elucidate patterns of correlation between bleaching and disease (*Lirman et al., 2014*). All of the ramets tracked had been at the nursery (common garden) for at least 3 years prior to the onset of bleaching.

We hypothesized that bleached ramets should be more susceptible to disease than their unbleached counterparts and that certain coral genets would have genetic pre-dispositions towards disease and bleaching resistance or susceptibility. Our results confirmed our hypothesis regarding the effect of genet. However, to our surprise, results revealed a significant negative correlation between bleaching and disease. These findings lead us to postulate a model whereby *Symbiodinium* may suppress host immunity. According to this theoretical framework, bleaching events may be associated with a transient increase in host immunological capacity, despite the nutritionally detrimental loss of *Symbiodinium*.

## MATERIALS AND METHODS

The strong El Niño Southern Oscillation (ENSO) event that occurred in 2014 triggered mass coral bleaching events and subsequent disease outbreaks in the Greater Caribbean and the Florida Reef Tract (*Manzello, 2015*; *Precht et al., 2016*; *Lewis et al., 2017*). Ramets of *A. cervicornis* propagated since 2007 within the *in situ* University of Miami "North Nursery" at N 25.488; W 80.109 were monitored by the same observer using SCUBA at four time points (September and November 2014, January and March 2015) under permits SAL-14-1086-SCRP, BISC-2014-SCI-0018, and BISC-2015-SCI-0018.

Within the nursery, multiple ramets belonging to the same genet grow on individual pedestals raised off of a common cement block. Each block containing clonal ramets belonging to the same genet rests on a sand bottom all within approximately 100 m of each other at an approximate depth of 7 m. No ramets were in physical contact for

the duration of the study. During the bleaching event and subsequent recovery, lasting from September 2014 through March 2015, 152 ramets representing 21 *A. cervicornis* genets were tracked. These genets were previously genotyped and identified as genetically distinct using microsatellite markers (*Baums, Miller & Hellberg, 2005*; *Baums et al., 2009*; *Lirman et al., 2014*). During every time point, each ramet was photographed and scored for presence or absence of bleaching using a calibrated colorimetric card as a reference (*Siebeck et al., 2006*). Any visible presence of disease was also recorded when an easily discernible linear boundary between apparently normally pigmented (tan to brown) tissue and transparent tissue and visible skeleton was observed. Each ramet was then assigned to one of the following categories based upon observations: "bleaching without disease", "bleaching with disease", "no bleaching without disease", or "no bleaching with disease". Manifestation of a tissue loss condition was noted as disease, because it followed a linear progression of tissue loss from the base progressing towards the tips in a manner similar to white band disease. However, our study did not fully explore the pathogenesis of this phenomenon and it should properly be referred to as a tissue loss disease. The individual history of one ramet throughout the entire duration of the study was considered the experimental unit, so that if a ramet bleached, recovered, and later experienced disease, it was grouped as "bleaching with disease" even though bleaching and tissue-loss conditions never co-occurred. A Fisher's exact test was employed to detect significant effect of genet on likelihood of bleaching or disease. To determine which genets were significantly different from each other, a Bonferroni corrected pair wise fisher's exact test was performed. A *chi*-squared test for independence was carried out to determine whether bleaching and disease were correlated or independent. Expected values were calculated for each category based upon the null hypothesis that bleaching and disease were fully independent as follows:

$O_{\%B}$ = observed % of ramets bleached

$O_{\%D}$ = observed % of ramets with disease

$O_{\%B\&D}$ = observed % of ramets bleached and diseased

Bleaching without disease = $(O_{\%B} - O_{\%B\&D}) \times$ total ramets

Disease without bleaching = $(O_{\%D} - O_{\%B\&D}) \times$ total ramets

Bleaching with disease = $O_{\%B} \times O_{\%D} \times$ total ramets

No Bleaching or disease = $(1 - (O_{\%B\&D} + O_{\%D} + O_{\%B})) \times$ total ramets.

Genets (9 of 21) that contained neither a bleached nor a diseased ramet over the entire duration of the study were removed from statistical analyses. We reasoned that these genets lacking vulnerability to both bleaching and disease are unsuitable for studying the interaction of bleaching and disease (*Vollmer & Kline, 2008*). Expected values for each test were calculated based upon the pool of observations inclusive of all those genets analyzed by each respective test. Only genets which showed neither bleaching nor disease in all of their ramets were removed from analyses.

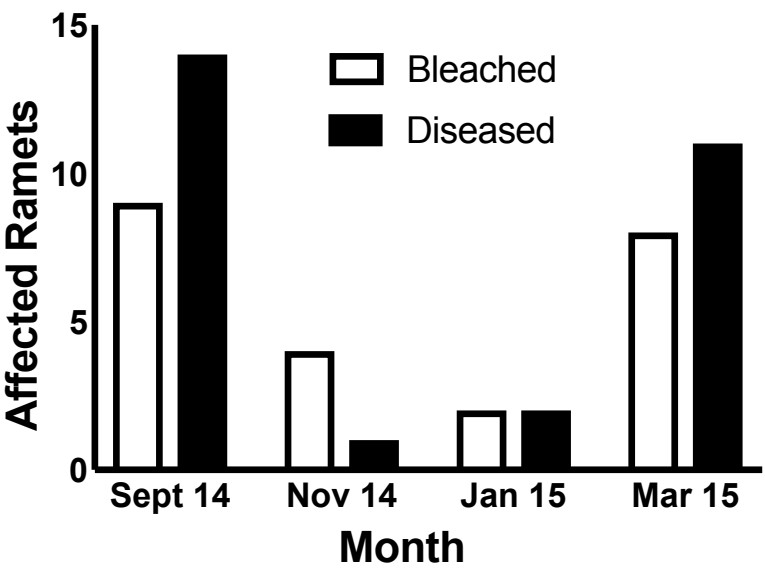

**Figure 1** Monthly prevalence of bleached and diseased coral ramets of *Acropora cervicornis* between September 2014 and March 2015 in the "North Nursery" at Biscayne Bay (N 25.488; W 80.109).

## RESULTS

In September 2014, nine ramets were bleached while fourteen ramets were affected by a white band-like tissue loss disease. In November 2014, four ramets were bleached, while one experienced tissue loss. In January 2014, two ramets were bleached and an additional two were afflicted by tissue loss disease. In March 2015, both bleaching and tissue loss disease increased in prevalence to eight and eleven cases, respectively (Fig. 1). During the entire period, 19 of the 152 (12.5%) *A. cervicornis* ramets showed signs of bleaching, while 28 ramets (18.4%) showed signs of this tissue loss disease. Only one ramet (0.7%) showed signs of both bleaching in Sept 2014 and disease recorded in March 2015, though pigmentation had recovered prior to the onset of disease. No ramet with simultaneous bleaching and disease was ever observed.

The tissue loss disease appeared to follow a linear progression from the base towards the apical tips of ramets in a manner reminiscent of white band disease (Fig. 2). However, molecular analyses necessary to confirm the identity of the disease were not conducted and the disease we observed is henceforth referred to as a "tissue loss disease". Furthermore, preliminary transmission trials bringing unaffected ramets into contact with the active lesions were unable to induce transmission although a linear progression of tissue loss was apparent. A highly significant negative correlation was detected between the presence of bleaching and disease ($\chi^2 = 7.14$, $p = 0.0075$). In total, nine of the 21 genets did not contain a single ramet suffering from either bleaching or disease during the monitoring period. There were significant differences between genets' proportion of ramets bleached,

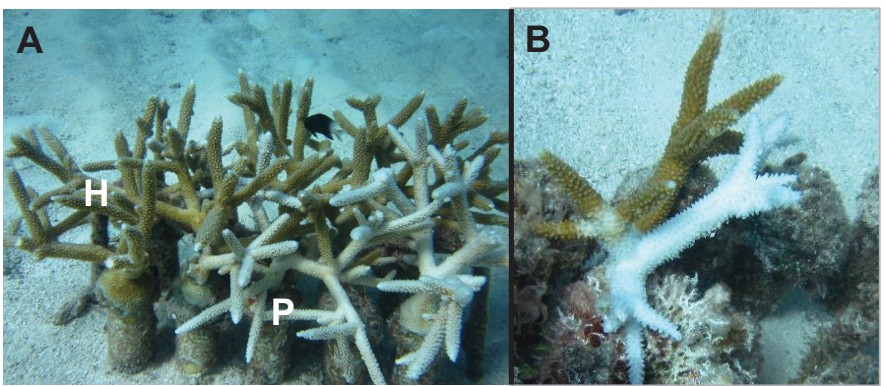

**Figure 2** **Images of bleaching and diseased colonies of *Acropora cervicornis* within the North Nursery.** Examples of bleaching and diseased colonies of *Acropora cervicornis* within the North Nursery. (A) Several ramets, some of which show normal, healthy pigmentation (H), while others are bleached pale (P). (B) One ramet showing signs of white band-like white syndrome. Photographs taken by Stephanie Schopmeyer.

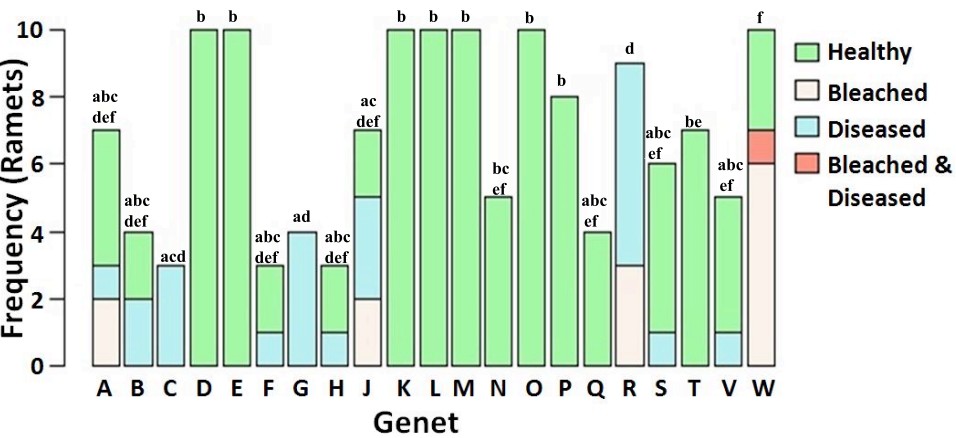

**Figure 3** **Frequency of health status in *Acropora cervicornis* corals as a function of genet identity.** Genet identity has a significant effect on the probability for each studied health status ($p < 0.0001$). Shared letters between genets indicate no significant difference. Both the trend for fewer significant comparisons for those genets with few ramets and low overall statistical power $1 - \beta = 0.31$ suggest that more significant differences could have been detected with a larger sample size.

diseased, neither bleached nor diseased, or both bleached and diseased despite limited statistical power to detect a medium sized effect ($p < 0.0001$, $1 - \beta = 0.31$, Fig. 3) (*Cohen, 1992*).

## DISCUSSION

Focusing on a ramet by ramet basis, our study revealed a negative correlation between bleaching and tissue loss disease during the thermal event in 2014 while simultaneously documenting a positive temporal relationship between bleaching and disease at a population scale (Fig. 1). While negative correlation between bleaching and a tissue loss disease has
been previously documented (*Bruno et al., 2007*), ours is the first report that shows a negative correlation between bleaching and a tissue loss disease in individually tracked ramets.

Previous work has indicated that coral genet identity influences diverse host phenotypes such as bleaching susceptibility, disease resistance, growth rate, and morphology (*Willis & Ayre, 1985*; *Vollmer & Kline, 2008*; *Bowden-Kerby & Carne, 2012*; *Lirman et al., 2014*). Our study further contributes to the body of genotypic response literature, suggesting that host genotype is a significant factor to consider for effective conservation and restoration. Unfortunately, the limited statistical power of our analyses may have contributed to our detection of relatively few significant comparisons (Fig. 3). A much larger analysis including 443 ramets equally distributed amongst genets would have provided a much greater statistical power of 0.90, but was beyond the scope of the current study.

Despite the great importance of coral host genetics in determining both bleaching and disease resistance, presence of genets susceptible to only one of the two conditions and resistant to the other does not appear to have solely driven the negative correlation between bleaching and disease. This is evident especially in genets A, J, R, and W (Fig. 3) which contain both ramets that suffered from disease and other ramets that suffered from bleaching, although never at the same time (recall that the ramet classified as bleached and diseased in genet W first bleached and became diseased only after recovering from bleaching). Such a pattern suggests that unbleached ramets within these genets later suffered increased disease susceptibility relative to their clones.

The role of *Symbiodinium* identity has also been strongly implicated in physiological response to bleaching and disease (*Baker, 2004*; *Tchernov et al., 2004*; *LaJeunesse et al., 2009*; *Silverstein, Cunning & Baker, 2014*; *Rouzé et al., 2016*). Results from a representative subset of samples taken from diverse genets indicate that no ramet had greater than a minimal (<2%) variance from exclusively hosting type A3 *Symbiodinium* (D Merselis et al., 2018, unpublished data). Therefore, we suggest that host genetics, not *Symbiodinium* identity is responsible for observed significant differences between genets. Not only do genet dependent differences in physiology inform which genets will do best in response to one or two focal stressors, but more importantly, which genets we are likely to lose. Given the precipitous decline of Caribbean reefs, and *Acropora cervicornis* in particular, we suggest that surviving genets likely posses anthropogenically robust traits, even if a study on any one given stressor indicates susceptibility. These differential responses should motivate not only the crossing of very bleaching or disease resistant genets, but also the inclusion of genets clearly at risk, but likely possessing unknown resistances to other anthropogenic stressors.

While increased sample size would have benefited analysis at the genet level and probably allowed for the detection of more significant differences ($1 - \beta = 0.31$), physiology must be studied at the level of the individual (in this case ramet). Without knowledge of the history of an individual gathered over multiple time points, it is not possible to ascertain whether an individual was not affected by bleaching or disease, suffered only bleaching, suffered only disease, or was afflicted by both bleaching and disease. When individuals are not tracked, but the prevalence of bleaching and tissue loss diseases are followed, it is clear

that disease and bleaching are linked through time during temperature anomalies. (Fig. 1 of this study; and *Muller et al., 2008*). However, because both bleaching and many tissue loss diseases are dependent upon temperature as a common stressor, it is expected that they should co-occur along spatial and temporal scales (Fig. 1), sharing high incidence where thermal stress has been severe and low incidence where thermal stress is mild (*Muller et al., 2008*). Without data to show that individuals and not just populations are first affected by bleaching and then disease, a physiological link cannot be supported. As exemplified here, when data is presented on an individual basis along a time series, it is possible that those individuals that bleach may be less prone to disease despite temporal co-occurrence within the population. We suggest it is possible that a negative correlation between bleaching and tissue loss diseases on an individual basis may have been overlooked by previous investigations, because individuals were not tracked across multiple time points (*Cróquer & Weil, 2009*). Monitoring individual corals (ramets) within a common garden nursery allowed us to control against co-occurrence of bleaching and tissue loss diseases as a result of spatial variation in environmental conditions while enabling repeated assessment of individuals with known bleaching and disease history.

A possible explanatory mechanism for a negative correlation between bleaching and tissue loss diseases may hinge upon the immuno-suppressive nature of intracellular symbioses. Intracellular parasites and mutualists modulate host immunological defenses in order to facilitate their intracellular lifestyles (*Oster, Kenyon & Pedersen, 1978*; *Fytrou et al., 2006*; *Douglas, Bouvaine & Russell, 2011*; *Ratzka, Gross & Feldhaar, 2012*; *Zheng, Tan & Xu, 2014*). Examples are diverse including Rickettsea (*Oster, Kenyon & Pedersen, 1978*), *Walbachea* (*Fytrou et al., 2006*), *Buchnera* (*Douglas, Bouvaine & Russell, 2011*), *Spiroplasma* (*Herren & Lemaitre, 2011*), *Sodalis*, *Wigglesworthia* (reviewed in *Ratzka, Gross & Feldhaar, 2012*), and *Plasmodium* (reviewed in *Zheng, Tan & Xu, 2014*), a distant relative of *Symbiodinium* that interferes with cellular processes to prevent apoptosis (*Kaushansky et al., 2013a*; *Kaushansky et al., 2013b*).

Further evidence is apparent within *Symbiodinium*—Cnidarian symbioses. Cnidarians hosting *Symbiodinium* express an altered distribution and expression of Rab proteins when compared to their apo-symbiotic con-specifics. This alternative regulation of Rab proteins preserves the symbiosis by preventing the maturation of the symbiosome, the vacuole where the symbiont resides, into a lysosome (*Chen et al., 2004*; *Riesgo et al., 2014*). This same dysregulation mechanism possibly impairs the ability for phagosomal degradation of pathogens by cnidarians hosting *Symbiodinium*. Further, apoptosis, an important immune response, is down-regulated in symbiotic *versus* aposymbiotic sea anemones (*Rodriguez-Lanetty, Phillips & Weis, 2006*; *Oakley et al., 2016*; *Matthews et al., 2017*; E Medrano et al., 2018, unpublished data), while potential cell adhesion markers facilitating pathogen entry are upregulated (*Rodriguez-Lanetty, Phillips & Weis, 2006*; *Yuyama, Watanabe & Takei, 2010*; *Riesgo et al., 2014*). This leads us to think that *Symbiodinium* containing host cells are immune-suppressed.

*Symbiodinium* may also promote immunological tolerance of their cnidarian hosts. Exogenous application of tolerogenic factors both decreases immune response of *Exaiptasia pallida* and prevents it from bleaching under elevated temperatures, while the treatment

with an anti-tolerogenic factor prevents symbiosis establishment and stimulates host immune function (*Detournay et al., 2012*; *Berthelier et al., 2017*). Recently, anthozoan TGFβ receptor and other modulators of immune response were proven to be regulated by *Symbiodinium* produced miRNAs *in hospite* (*Baumgarten et al., 2017*). Likewise, many immunological processes lead to the generation of ROS, a primary trigger of coral bleaching (*Lesser, 1996*), suggesting that corals with high capacity for immunological response may be more susceptible to bleaching. Therefore, corals with the highest immunological activity at the onset of thermal stress may be at an elevated risk of bleaching ( *Brandt & McManus, 2009*). A host previously lacking in *Symbiodinium* may be better prepared to confront invading pathogenic microbes. Conversely, corals better prepared to confront invading microbes may be more likely to expel their symbionts as a side effect of an immune response.

Under our proposed model (see Fig. 4), bleaching corals gain a transient immunological advantage as a result of shedding their symbionts. Despite disparate thermal bleaching thresholds both between and amongst species, genotypes, and geographic locations, little is known about the "trade-off" or ecological cost for increased bleaching resistance , although a slower growth rate for bleaching resistant genotypes has been supported for *Acropora cervicornis* (*Ladd et al., 2017*). We suggest that our proposed model is a trade-off of decreased bleaching resistance in exchange for enhanced immunological function and vise versa.

It is important to note that immunological responses are metabolically costly. Bleaching reduces or completely stops the assimilation of *Symbiodinium* derived nutrition. Therefore, the immunological capacity of a bleached coral would eventually be hindered by decreasing energetic reserves. The model, which assumes that *Symbiodinium* density is directly related to immune-suppression, illustrates that corals would have evolved to consider immunological capacity when setting a bleaching threshold, alongside tolerance for oxidative and thermal stress. By setting a high bleaching threshold, corals forego the putative immunological advantages of bleaching, but retain *Symbiodinium* until their antioxidant protections against thermal stress become overwhelmed. This strategy would maintain higher energetic reserves and may prove more successful under long term thermal stress scenarios where energetic reserves may become limiting to the maintenance of homeostatic processes and immunological capacity. Conversely, corals may set a low bleaching threshold to protect against infectious disease in the short term at the risk of susceptibility to starvation if thermal stress is long term and prevents the re-population of *Symbiodinium*. Corals exploiting this latter strategy suffer ongoing pathogen exposures or thermal stress and may be forced to recover symbiont populations in order to prevent starvation. Resultantly, these corals simultaneously suffer the onset of *Symbiodinium* mediated immunological suppression and depleted energetic stores. In congruence with field observations, this is perhaps why disease outbreaks may intensify upon the onset of bleaching recovery (*Brandt & McManus, 2009*). In further agreement with field observations, those corals which bleach and are still unable to prevent the onset of disease outbreaks would be expected to suffer the greatest tissue loss (*Muller et al., 2008*; *Brandt & McManus, 2009*). These corals lack both the immunological competency to prevent infectious disease and the

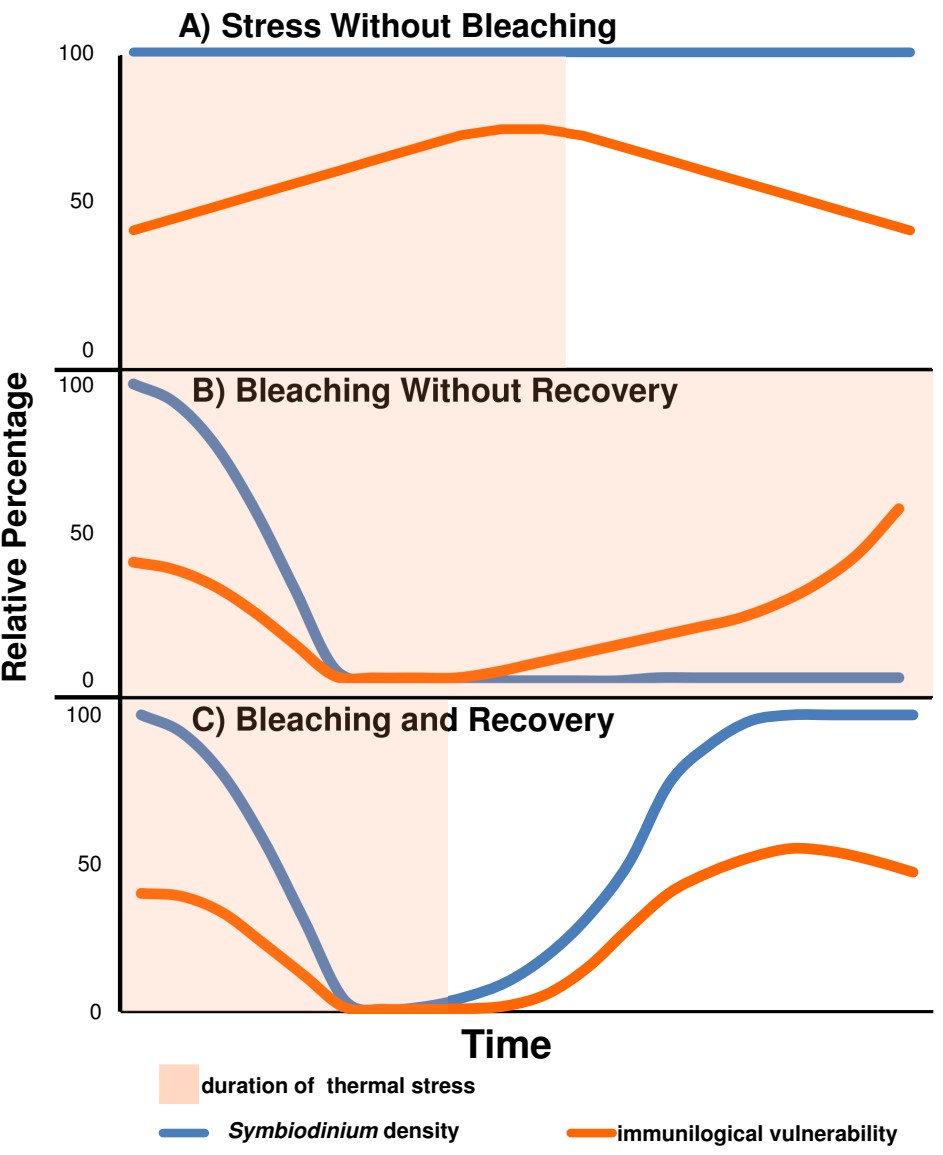

**Figure 4  Predicted relative immunological vulnerability assuming *Symbiodinium* have an immuno-suppressive affect on the coral host.** (A) A coral which does not bleach in response to a stress event may see immunological vulnerability increase until the stress event subsides. (B) A coral which bleaches and does not recover is free from symbiont immunosuppression, but eventually becomes immunologically vulnerable because of energy store deprivation. (C) A coral which bleaches and recovers may minimize *Symbiodinium* immunosuppression in the short term, but suffer from reduced energy stores and immuno-suppression of returning *Symbiodinium* upon recovery.

energetic stores to mount a sustained response. Also in agreement with field observations, *Symbiodinium* densities are greater on eutrophied reefs which also have greater tissue loss disease prevalence (*Muscatine et al., 1989*; *Shantz & Burkepile, 2014*; *Vega Thurber et al., 2014*). It is worth noting that reefs with more frequent thermal anomaly are known both for their bleaching resistance and white syndrome susceptibility, although it should also be noted that they are more susceptible to brown spot disease (*Hume et al., 2013*; *Fine, Gildor & Genin, 2013*; *Palumbi et al., 2014*; *Randall et al., 2014*).

Here, in addition to proposing a new model for infectious disease susceptibility in the context of coral bleaching, we establish a testable hypothesis: Coral bleaching confers a transient immunological advantage to the coral host. While the present study's sample size is limited and canonical logic has historically supported a causal and positive relationship between bleaching and tissue loss diseases, our proposed hypothesis is supported by molecular work and alternative interpretations of several field studies. Further testing is warranted, especially as reefs are exposed to increasingly frequent and intense thermal anomalies. Beyond our proposed hypothesis, this study adds further support to numerous works demonstrating the importance of coral host genotype in determination of diverse physiological traits.

## ACKNOWLEDGEMENTS

The authors thank Dr. Kuulei Rodgers, Dr. Tanya Brown, Dr. Anthony Bellantuono, Ms. Ellen Dow, Ms. Cindy Lewis, Mr. Daniel Quintero, and two anonymous reviewers whose comments improved this manuscript, Stephanie Schopmeyer, Crawford Drury, Dalton Hesley, and Patricia Waikel for diving logistical support, and Dr. Wensong Wu for statistical advice.

### Funding

This research was funded by the National Science Foundation (NSF-OCE-1503483 and NSF-IOS-1453519). The funders had no role in study design, data collection and analysis, decision to publish, or preparation of the manuscript.

### Grant Disclosures

The following grant information was disclosed by the authors:
National Science Foundation: NSF-OCE-1503483, NSF-IOS-1453519.

### Competing Interests

Mauricio Rodriguez-Lanetty is an Academic Editor for PeerJ.

### Author Contributions

- Daniel G. Merselis conceived and designed the experiments, performed the experiments, analyzed the data, contributed reagents/materials/analysis tools, prepared figures and/or tables, authored or reviewed drafts of the paper, approved the final draft.

- Diego Lirman conceived and designed the experiments, contributed reagents/materials/analysis tools, authored or reviewed drafts of the paper, approved the final draft.
- Mauricio Rodriguez-Lanetty conceived and designed the experiments, contributed reagents/materials/analysis tools, prepared figures and/or tables, authored or reviewed drafts of the paper, approved the final draft.

### Field Study Permissions

The following information was supplied relating to field study approvals (i.e., approving body and any reference numbers):

Field experiments were approved by the Fish and Wildlife Conservation Commission.

### Data Availability

Dryad DOI: 10.5061/dryad.d8sv77t.

### Supplemental Information

Supplemental information for this article can be found online at http://dx.doi.org/10.7717/peerj.4494#supplemental-information.

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
