# Peer review of "Symbiotic immuno-suppression: is disease susceptibility the price of bleaching resistance?"

_PeerJ, doi:10.7717/peerj.4494_

## Round 0.1 · original submission · Major Revisions

I am sorry for the delay in getting this back to you, but we needed to get an additional review when the first two came back diametric in their evaluations, with one recommending minor revisions and the other a complete rejection. The third referee came back much more similar to the first, with a recommendation of minor revisions as well. Given that 2 of the 3 referees are supportive of the publication of the work, I am returning a decision for revision to allow the authors to revise the manuscript in response to the feedback of these 3 experts in the field. I can see this subject is controversial, and I find myself leaning in favor of having the work in the literature to promote additional study that will determine which of the referee opinions is more accurate. However, I urge the authors to take the feedback of the most critical referee seriously and undertake a relatively major revision, because this person has some good points, and clearly has a negative opinion of the work as presented. If it were me, I would want to take the feedback and address it directly in the manuscript, because if 1/3 of your readers have a similar reaction to your manuscript, it will have less impact than you would hope. Still, given the response of the two others, I am willing to consider revisions from the authors at whatever level you feel is appropriate so long as you correct the incorrect nomenclature for disease studies as highlighted by the second referee. I look forward to seeing your revisions, and am happy to discuss the reviews in person if you would like additional guidance on my thoughts for revision based on the feedback from these referees.

·

Basic reporting

meets all standards

Experimental design

Experimental design is solid just a few small suggestions.

Was there error associated with observer variability? Did the same observer assess bleaching and disease across all time periods? If so it should be mentioned in the methodology.

12.5% showed signs of bleaching and 18.4% of disease. This is a small percentage of the total sample size. The authors recognize the benefit of increasing the sample size on lines 180-181 but do not give details of what sample size would be appropriate. A power analysis of the data should give the suitable sample size.

Validity of the findings

Fig. 2 shows a number of fragments that appear intertwined and touching each other. Was a spatial analysis conducted on whether or not disease increased with decreased distance from a diseased fragment? There is no mention as to how they were oriented within the nursery or whether or not direct contact can infect a white band-like white syndrome.
In lines 54-60 there is minimal discussion of transmission. A more thorough discussion of vectors and transmission in the introduction would be helpful that also includes how you are defining “disease”. You refer to the correlation between bleaching and “disease” but is this white band-like white syndrome really a disease or is it a syndrome or other condition? This should be made clear in the beginning and in the methodology. Your terminology is very important.

Additional comments

Overall the paper is well written and in-depth. I believe this research significantly contributes to the larger field of climate change research.
The authors include plausible explanations with detailed support of symbiont immune suppression to back their findings of a negative correlation between bleaching and disease.
Tracking individuals with a known prior history over time helps determine the link between bleaching and disease. The authors make a good case for the limitations of other field studies that omit this.

Reviewer 2 ·

Basic reporting

The manuscript was ok on this component although I do make recommendations for changes to the way the results are presented.

Experimental design

The strength of the study is in the following of individual ramets through time. However, the limited number of ramets which displayed bleaching or tissue loss disease was an extremely small proportion of the overall community thus drawing conclusion based upon this data difficult. It would have also been much more informative if the zooxanthellae clade of the genets had been reported as zooxanthellae clade has been shown to affect bleaching and disease susceptibility. Without that information conclusions drawn are limited.

Validity of the findings

Conclusions drawn and the hypothesis of immune-suppression by zooxanthellae are not supported by the data or other works. They don't address any other factors that might produce differential bleaching and disease susceptibility with a major failure being not linking zooxanthellae clade to the findings. They also do not report their findings in a way that allows the study to be evaluated properly. What size were all of these ramets? all similar sizes? clarify. They also need to report genets/ramets relationships. Were the ramets that bleached always from the same genet? Or did that vary among the different time periods? Ditto for the ramets displaying disease. There is also a current problem in many of the Florida nurseries with fireworm predation common and commonly mistaken for disease. They may have evidence (clear linear progression? nocturnal search for fireworms? etc) that it was disease not predation but that was not presented. Although it is great that they followed individual ramets through time, the study as presented does not support their conclusions.

Additional comments

Looking at individual ramets through time with temperature is a good idea but the authors jumped to too many conclusions based on many false assumptions. They also use incorrect nomenclature for disease studies. I have added suggestion to the text to help improve any future submissions.

Annotated reviews are not available for download in order to protect the identity of reviewers who chose to remain anonymous.

Reviewer 3 ·

Basic reporting

In general I think that this is a good first study of individual genets to understand the relationship between bleaching and disease outbreak.

Refs are sufficient, but there are a few areas that I suggest having added references.
Line 77: "...enhanced by thermal stress (ADD REF)"


Raw data is shared, article structure and figs are professiona.

Hypothesis was stated and tested.

Experimental design

The authors acknowledge that their sample size is low, and this is due to it being a field study of corals from a nursery. A few things need to be clarified about this study:

1. Are the genets known genotypes that have disease or bleaching resistance? Lirman Lab has done studies on this, and I wonder if some of the results could be due to the genotypes that are being studied.

2. More detail on the genotypes studies would greatly help this paper be stronger.

Validity of the findings

The novelty was assess and I agree with the authors that this is a very important first step in understanding this phenomena. Data, stats and conclusions were sound and concluding figure was very informative as a model for future studies.

Additional comments

Overall great paper. There are a few spelling, grammar issues and questions about things:

line 66: replace dispute with debate.
line 91: "these observations these disease outbreaks" doesn't make sense.
line 100: explain here what you mean by colonies...you later bring it up, but it is confusing because there is a lot of switching between coral, coral colony and genet...which I think you mean as the same thing, but it confusing for the reader.
line 101: in-situ should be in situ (in italics)----this happens through out the manuscript.
line 125: "...each coral..." again what are you talking about here (genet or colony or ramet?)

Lines 141-143: I suggest putting these first, then showing the equations.

---

## Round 0.2 · accepted · Accept

Thanks for your thoughtful revision. Each of the referees are now supportive of the acceptance of your manuscript, so I am happy to move it along into production.

·

Basic reporting

meets all standards

Experimental design

The experimental design is solid. Authors adequately addressed all concerns regarding observer variability and sample size. A power analysis was conducted and discussed in text.

Validity of the findings

Manuscript was revised to address comments concerning validity. Concerns about transmission of disease from ramets in close proximity to one another was allayed in the revision.

Additional comments

The authors did their due diligence in addressing each comment thoroughly in the rebuttal.

Reviewer 2 ·

Basic reporting

Much improved.

Experimental design

Good

Validity of the findings

Much improved

Additional comments

Fig 3 really adds to understanding this study! Please change the word "white band disease" on line 101 to "white syndrome" . White band disease does not occur on the GBR. There is a spelling error on line 293 , posses.

Nice job with the revisions!